# Outer Membrane Vesicles: An Emerging Vaccine Platform

**DOI:** 10.3390/vaccines10101578

**Published:** 2022-09-21

**Authors:** Dharmendra Kashyap, Mrutyunjaya Panda, Budhadev Baral, Nidhi Varshney, Sajitha R, Vasundhra Bhandari, Hamendra Singh Parmar, Amit Prasad, Hem Chandra Jha

**Affiliations:** 1Department of Biosciences and Biomedical Engineering, Indian Institute of Technology Indore, Indore 453552, India; 2Department of Life Science, National Institute of Technology Rourkela, Rourkela 769008, India; 3Amity Institute of Biotechnology, Amity University Noida, Amity 201313, India; 4Department of Biological Science, National Institute of Pharmaceutical Education and Research, Hyderabad 500037, India; 5School of Biotechnology, Devi Ahilya Vishwavidyalaya Indore, Indore 452001, India; 6School of Basic Sciences, Indian Institute of Technology Mandi, Mandi 175005, India

**Keywords:** outer membrane vesicles (OMV), vaccines, adjuvants

## Abstract

Vaccine adjuvants are substances that improve the immune capacity of a recombinant vaccine to a great extent and have been in use since the early 1900s; they are primarily short-lived and initiate antigen activity, mainly an inflammatory response. With the developing technologies and innovation, early options such as alum were modified, yet the inorganic nature of major vaccine adjuvants caused several side effects. Outer membrane vesicles, which respond to the stressed environment, are small nano-sized particles secreted by gram-negative bacteria. The secretory nature of OMV gives us many benefits in terms of infection bioengineering. This article aims to provide a detailed overview of bacteria’s outer membrane vesicles (OMV) and their potential usage as adjuvants in making OMV-based vaccines. The OMV adjuvant-based vaccines can be a great benefactor, and there are ongoing trials for formulating OMV adjuvant-based vaccines for SARS-CoV-2. This study emphasizes engineering the OMVs to develop better versions for safety purposes. This article will also provide a gist about the advantages and disadvantages of such vaccines, along with other aspects.

## 1. Introduction

OMVs are outer membrane vesicles of gram-negative bacteria, having stellar intrinsic immunostimulatory properties; they are nano-sized particles having a mean size between 20 to 200 nm; they are spherical buds of the outer membrane filled with periplasm and are usually bilayered. The bilayered nature of OMVs protects the lumen from immediate degradation by extracellular enzymes. OMVs can fuse with other cells allowing for the transfer of lumen contents and, therefore, providing a host-pathogen interaction. OMVs have several biological functions, including the delivery of proteins and toxins to target cells, transport of various effectors between bacterial cells, protection of nucleic acid during intercellular transport, and bacterial defense [1]. Several bacterial species produce OMVs, such as *Escherichia coli*, *Pseudomonas aeruginosa*, *Shigella* sp., *Salmonella* sp., *Helicobacter pylori (H. pylori)*, *Campylobacter jejuni*, *Borrelia burgdorferi*, *Vibrio* sp., *Neisseria* sp. [2]. For the first time in 1959, it was observed that the cell-free filtrate of *V. cholera* induces an immune response in rabbits. Then in 1966, secretion of lipopolysaccharide in the form of spherical-shaped pouches was observed in *E. coli* culture grown on a lysine-limiting medium [3]. Indian Scientists Smriti Narayan Chatterjee and J. Das 1966–67 used Transmission-electron microscopy and discovered OMVs in *V. Cholerae* as membrane-bound vesicles produced by blebbing out and pinching off of outer membranes [4]; these OMVs have been alternatively termed “microvesicles”/“outer membrane fragments”, “blebs”, and “extracellular vesicles” [5].

Analysis of OMVs shows the presence of various outer membranes (OmpA, OmpC, and OmpF) and periplasmic proteins (AcrA and alkaline phosphatase). Several virulence factors are also present, which aid in the adhesion and invasion of host tissue. Periplasmic proteins present in the inner leaflet of the outer membrane show increased incorporation within OMVs compared to the tightly bound proteins of the inner membrane [2]. It was demonstrated that the lipid composition, SDS page protein profile, and specific activities of several membrane enzymes were similar in both the membrane vesicles and the outer membrane. Moreover, proteins including lipoproteins were less abundant in OMVs than in the outer membrane, indicating the likely origin of OMVs from specific outer membrane regions [6]. Multiple proteomic analyses reveal that inner membrane proteins and protoplasmic proteins are also abundant in OMVs. A recent study showed that OMVs are likely formed by cell lysis (The cell wall is degraded by endolysin, triggering explosive cell lysis, allowing for the fragmented membranes to round up forming OMVs), explaining the presence of inner membrane and cytoplasmic contents in OMVs [7]. Preferential packaging of proteins to OMVs is regulated by the total protein content of OMVs. Enrichment of proteins to OMVs takes place when protein contents of OMV are significantly higher with respect to the cellular concentration [8].

Nonetheless, lipids are integral structural components of OMVs. Just as in proteins, there are occurrences of lipids in the outer membrane but not in OMVs. Glycerophospholipids, phosphatidylglycerol, phosphatidylethanolamine, and cardiolipin are all important lipids present in OMVs and are associated with their curvature. Notably, a higher proportion of fatty acids results in their rigid structure [2]. OMVs carry luminal and surface-associated DNA, and a clear distinction is observed during DNase treatment; luminal DNA is present even after treatment compared to surface-associated DNA. Besides DNA, RNA, plasmid; and chromosomal DNA;phage DNA is also present in OMVs. Similar to protein incorporation, DNA is thought to be incorporated into OMVs after cell lysis during biogenesis [2] (Figure 1).

## 2. Formation of OMVs

There are three major models behind the formation of OMVs. The first is based on the loss or relocation of covalent linkages between the peptidoglycan layer and the outer membrane; as lipoprotein is associated with the linkage, it is hypothesized that a mutation in the lipoprotein gene could lead to potential hyper vesiculation. The missing cross-links along with an outer membrane grow faster, allowing the outer membrane to protrude and initiate vesiculation. In the second model, peptidoglycan fragments, misfolded proteins protrude into the periplasmic space, which exerts a turgor pressure on the outer membrane leading to the pinching off of OMVs. Another lipoprotein, vfgl, contributes to an increased peptidoglycan production or downregulation of transglycosylases, which again leads to an increased turgor pressure on the outer membrane, which then pinches off OMVs to decrease the pressure exerted [2]. The enrichment of the curvature-inducing molecules such as B-band lipopolysaccharide and the quinolone PQS of *Pseudomonas aeruginosa* forms the basis of the third model of OMV production. PQS is thought to increase anionic repulsions between lipopolysaccharide molecules leading to membrane budding due to isolated divalent cations [9]. Hydrophobic molecules such as PQS allow the formation of OMVs by inducing the outer membrane curvature, thus allowing the spread of these hydrophobic molecules in a hydrophilic environment. OMVs allow soluble proteins to be encased in an insoluble protective membrane sheath, often enabling the transport of these proteins in a hostile environment (having high temperatures or degrading enzymes such as proteases). Moreover, encasing such soluble proteins allows them to travel a greater distance while maintaining their concentration [10].

Another mechanism by which PQS causes blebbing is the asymmetric expansion of the outer layer of the outer membrane. However, this model is restricted as it is species-specific [9]. OMV-mediated secretion involves the presence of an insoluble membrane surrounding the soluble material. It can be regulated both temporally and spatially. In pathogenic bacteria, proteins known as adhesins present in the outer membrane, are crucial for colonization of the host tissue as they cause coaggregation. OMVs present multivalent complexes of membrane adhesins [10]. The bacterial interaction with the host triggers the release of OMVs containing adhesin proteins, which promote the adhesion of the bacterial cells to the epithelial linings of the host [2]. In addition to reaching a particular site, OMVs also have the advantage of targeting a particular site via the binding proteins present on the surface of the OMVs. Three different types of OMV delivery have been known. First is the targeted lysis, thus releasing the contents of the vesicles. Second is the attachment and subsequent fusion of the membrane with another cell allowing the lumen contents to be transferred; this mechanism carries importance regarding horizontal gene transfer between the bacterial strains and occasionally among bacterial species. Thus, having an important role to play in evolution and population dynamics. Another delivery mechanism is endocytosis within eukaryotic cells, which results in a display of bacterial epitopes to the immune system [10].

These delivery mechanisms are also involved in delivering toxins-enriched OMVs to targeted cells, thereby increasing the pathogenicity of bacteria [2]. Under environmentally stressful conditions, the bacterium can pinch off membrane parts under attack, thereby preventing cell death. Surface attacking agents such as a lytic phage can quickly be removed using OMVs before the DNA is released into the cell; thus, helping to prevent infection. By employing a similar mechanism, bacteria can also escape the attack of antibiotics. Increased production of OMVs is also observed with mutated stress response genes. Stressors can influence both vesiculation levels and the content of the OMVs [10].

Packaging of enzymes such as proteases, glycosidases, and in rare cases, metal ions play an important role in the nutrient acquisition of bacteria. Moreover, OMVs undergo lysis in the proximity of the bacteria, allowing for the uptake of nutrients by the bacteria. In many environments, metal ions become a limiting factor for colony growth; thus, bacterial strains can prosper by absorbing the essential metal ions. Phosphoenolpyruvate, a catalytic product of OMV-carrying enolase, allows for the conversion of plasminogen to plasmin and colonization of the host tissue by degradation of the matrix proteins [2]. OMVs moderate biofilm growth, which is produced in response to stress, releasing exopolysaccharides that increase cell aggregation. OMVs increase the survivability of the cells by acting as a decoy with antibacterial agents’ vesicle-associated multidrug efflux pumps providing temporary refuge for the bacteria in the surrounding environment with OMV-aided formation biofilm [2] (Figure 2).

## 3. Species Producing OMVs

### 3.1. Helicobacter pylori

*Helicobacter pylori* (*H. pylori*), is a helical-shaped, microaerophilic, gram-negative bacteria [11]. It is catalase, urease, and oxidase-positive and contains 3–5 polar flagella for motility. *H. pylori* can colonize the stomach’s highly acidic environment by converting urea into ammonia with the help of urease; this ammonia neutralizes the acidic environment making it more hospitable for the bacterium to survive. It has evolved ways to interfere with the host’s immune responses, making it ineffective in eliminating the bacterium. Furthermore, their helical shape burrows into the stomach’s mucus lining and develops the infection. Usually, there are no symptoms of *H. pylori* infection, but it sometimes causes gastritis (inflammation of the stomach lining) or stomach ulcers. The infection for a longer time increases the risk of gastric cancer development [12]. *H. pylori* are observed to release OMV in both in vivo and in vitro conditions. OMV of *H. pylori* consists of phosphatidylglycerol (PG), phosphatidylcholine (PC), lysophosphatidylcholine (LPC), phosphatidylethanolamine (PE), lysophosphatidylethanolamine (LPE), cardiolipin, and lipopolysaccharide (LPS) [13]. Proteomic analysis suggests the presence of an allelic form of vacuolating cytotoxin (VacA), several porins and lipoprotein 20 (Lpp20), various adhesions such as sialic acid-binding adhesion (SabA), blood group antigen-binding adhesin (BabA), adherence-associated lipoprotein A (AlpA), OipA, adherence associated lipoprotein (AlpA), *H. pylori* neutrophil-activating protein (HP-NAP), urease, and some other associated virulence factors [14,15].

The cytotoxin-associated protein (CagA) is present on the surface of *H. pylori* OMVs; CagA is a virulence factor encoded inside the 40-kb *cag* pathogenicity island (PAI); this PAI also contains a type-IV secretion system that delivers CagA protein into the host cell. *H. pylori* neutrophil-activating protein (HP-NAP) is chemotactic for monocytes and neutrophils of humans and stimulates the cell to produce reactive oxygen species (ROS); ROS damages the gastric epithelium by releasing nutrients to the bacterium; therefore, delivering HP-NAP through OMV to gastric mucosa can increase the availability of nutrients to bacteria via ROS-mediated mucosal damage [11]. *H. pylori* OMV can trigger serum IgG response that contains antibodies having specificity for bacterial Lewis’s epitopes. Continuous release of OMVs in the cellular milieu contributes to the persistence of *H. pylori* in the stomach through maintaining biofilm. OMV of *H. pylori* contributes to chronic inflammation by delivering certain virulence factors and toxins to the gastric epithelium; they also deliver PE, LPS, and porins to the gastric epithelium, which respond by secreting IL-8 [16]. *H. pylori* OMV can induce apoptosis of gastric epithelium cells in a mitochondrial-independent and caspase-dependent pathway. It is observed that OMVs are internalized into the AGS cells in, which OMV-mediated delivery of peptidoglycan to cytosolic NOD-1 leads to the activation of NFkB and IL-8, this cytokine is acting as a chemoattractant to involve more neutrophils in the site of production; these *H. pylori* OMVs also regulated the proliferation of gastric epithelium cells, resulting in growth arrest and decreased cell viability. Once these OMVs are inside AGS cells, the VacA-mediated alkalization of late endosomal compartments leads to increased cytoplasmic iron levels and a decrease in GHS [11] (Figure 3).

### 3.2. Neisseria meningitidis

*Neisseria meningitides* (*N. meningitidis*) is a gram-negative bacterium that causes severe disease in humans called meningococcal meningitis. It is an inflammation of the meninges, which are membranous coverings around the brain and spinal cord. The most common symptoms of meningitis include high fever, stiff neck, headache, vomiting, confusion, and sensitivity to light. The bacteria are transmitted through respiratory droplets from carriers. 12 serogroups have been identified in *N. meningitides*; among them, six major pathogenic serogroups can cause an epidemic, they are A, B, C, W, X, and Y (WHO). Vaccines developed using capsular polysaccharides of the pathogen for serogroup A and C have been in use since the 1960s and for serogroups A, C, W, X, and Y since the 1980s [18]; these capsular polysaccharide structures have high immunogenic properties and are conjugated with carrier proteins that induce a high immune response. However, this capsular polysaccharide vaccine approach is not suitable for serogroup B of *N. meningitidis* due to the low immunogenicity and the risk of autoimmune response due to structural homology of this capsular polysaccharide structure with polysialylated form of neural cell adhesion molecule (PSA-NCAM) present in brain tissue of fetus. Therefore, the vaccine approach for *N. meningitidis* serogroup B has been broadly focused on outer membrane vesicles proteins (OMVp) [19].

OMVs are shed from outer membranes of *N. meningitidis* as vesicles containing lipopolysaccharide (LPS) and five major classes of OMVp: class 1 (PorA porin), class 2/3 (PorB porin), class 4 (Rmp), and class 5 (Opa) [20]. The native OMVs (nOMV) contain high amounts of lipopolysaccharide; therefore, they are treated with detergent sodium deoxycholate. Despite many OMP, the dominant antibody response is directed against Por A porin. However, this response is age-dependent, as studied in Chile using two OMV vaccines and found that the vaccines were less effective in infants. The detergent extraction process reduces the toxicity and LPS but, it also removes lipoproteins such as H-binding protein, which contributes to immunogenicity. An alternative method is using detergent-free OMVs produced by a meningococcal strain with genetically attenuated LPS (mutation in the lpxL1 gene leading to Penta-acylated lipid A with reduced endotoxicity). OMV vaccines for serogroup B of *N. meningitides* (VA-MENGOC-BC) are developed and tested by Finlay Institute, Cuba, during an epidemic between 1987 and 1989. It was shown to be 83% effective in young adults over 16 months. MenBvac was developed and tested by the Norwegian Institute of Public Health during an epidemic between 1988 and 1991. It was shown to be 87% effective in young adults over ten months. MeNZB was developed in partnership with WHO, New Zealand Government, the Norwegian Institute of Public Health, the University of Auckland, and Chiron between 2004 and 2008. It was shown to be 73% effective in young adults. Bexsero vaccine is the latest vaccine formed by the combination of detergent-extracted OMVs and three recombinant antigens from the New Zealand strain; this vaccine was developed by Novartis and licensed by the European Medicines Agency [18].

### 3.3. Campylobacter jejuni

*Campylobacter jejuni* (*C. jejuni*) is a spiral-shaped, rod-shaped, S-shaped, or curved-shaped gram-negative bacterium that is a significant cause of acute food-borne gastroenteritis found in animals; pigs, cattle, poultry, cats, and dogs. Clinical symptoms of the infection include fever, headache, nausea, abdominal pain, non-inflammatory diarrhea to inflammatory diarrhea with blood. It rarely causes death, mostly confined to infants, elderly persons, or patients with other severe diseases (WHO). The synchronized delivery of virulence factors such as adherence, the ability of invasion, production, and motility of toxins is a general mechanism through, which bacterial pathogens interact and cause damage to host cells and increase their survival rate. The mechanism of secretion of bacterial proteins and non-protein substances directly into the host cytoplasm requires direct contact of enteric bacteria with the host cells. Bacterial toxin such as cytolethal distending toxin (CDT) has been considered an essential factor for the pathogenesis of infection by *C. jejuni*; these CDTs belong to AB2-type toxins, consisting of three subunits-Cdt A, Cdt B, Cdt C [21]. Cdt A and C are binding proteins that deliver catalytic subunit Cdt B into the host cell; this Cdt B showcases DNase I-like activity and causes double-stranded DNA damage; this leads to cell cycle arrest at the G2/M stage and activation of DNA repair response [22].

Nonetheless, *C. jejuni* lacks most virulence-associated with the secretory system (except type III and type IV), it requires OMVs to deliver the proteins and non-protein substances into the host cell cytoplasm. Bacterial OMVs associated virulence factors regulate the host immune response, inducing IL-8, IL-6, TNF-, and hBD-3 and promoting inflammation or triggering of clearance of pathogens [23]. Proteomic profiling of OMV of *C. jejuni* suggests that 11 glycoproteins are also found in OMVs. Among them is a surface-exposed CjaA, well known for its antigenic properties and glycosylation related to virulence. Some periplasmic and transmembrane N-glycoproteins are components of OMV in *C. jejuni* [24].

## 4. OMV-Based Vaccine Delivery

Host immune responses are the first line of barrier to stopping any infection, but occasionally immoderate responses could lead to lethargic tissue damage. Thus, there is a need for a system that avoids excessive immune responses. Along with the adjuvant properties, OMVs also provide complete immunity as they carry the antigen of pathogens. Moreover, the non-replicative nature of OMVs makes them advantageous for antigen delivery to the host thus denying any fright of infection associated with whole cell vaccine against disease-causing pathogens. OMVs stimulate the innate immune system of the host via the activation of TLRs and NLRs as they contain various PAMPs such as lipoproteins, LPS, and pathogenic DNA fragments [25]. Studies have shown that OMVs with attenuated endotoxicity provide enhanced mucosal adjuvant properties in comparison to intranasal vaccines adjuvanted with cholera toxins; these OMVs are less reactogenic in comparison to native ones and also resulted in an increase in the production of IgG and IgA [19].

Nonetheless, OMVs play an important role in the delivery of various virulence factors to host cells, which initiate the host immune response. Therefore, OMVs-based delivery systems may be utilized to educate the immune system to counter pathogens in a con-trolled manner. Studies by Jang, Elmi, and Bauman showed that OMVs induce the host immune responses via interacting with the host epithelial cells [16,23,24]. Another study also revealed that the OMV of *H. pylori* carrying CagA is sufficient to evoke the immune response of gastric epithelial cells in place of bacteria [16,25]. Moreover, OMVs from Campylobacter jejuni contains outer membrane proteins that also potentially induce host immune response in the intestine [23]. Nonetheless, various pathogens also evoke a proinflammatory response viz; *P. aeruginosa* cytokines and chemokines production in lung epithelial, and macrophage cells in mice and rodents respectively. OMVs of other bacteria such as *N. meningitidis*, *L. pneumophila*, and *H. pylori*, *Salmonella* spp. enhance the proinflammatory response through the production of tumor necrosis factor-alpha (TNF-α), Interleukin-1 beta (IL-1β); chemokines (CXCL8, CCL3, and CCL4) and degranulation of eosinophils [26,27,28]. 

OMVs interact with dendritic cells and activate innate and adaptive immunity. OMVs have the potential to interact with a variety of immune cells, hence provoking proinflammatory responses. OMVs have enormous potential as endogenous vaccine candidates. Since OMVs carry surface antigens identical to that of bacterial surfaces it limits the ability of pathogens to mutate all the target antigens present on the vaccine thereby decreasing its ability to generate vaccine-resistant mutants. Moreover, OMVs in their native state possess many immunogenic properties, such as self-adjuvation and uptake by immune cells; this makes them an excellent autologous cell-derived vaccine delivery plat-form [29]. Bioengineered OMV-based vaccines can be developed with increased yields and reduced endotoxicity decorated with homologous and heterologous antigens to treat specific diseases. A versatile OMV-based vac-cine platform has been developed to elicit an anti-tumor immune response by presenting the antigens onto the OMV surface. Decorated with different protein catchers, the OMVs were able to display distinct tumor antigens. An effective tumor vaccine vector displaying neoantigens was thus developed [30].

### MERS-CoV and SARS-CoV-2

Middle East Respiratory Syndrome–related Coronavirus (MERS-CoV) is a zoonotic virus transmitted between animals and humans (WHO). It is a species of the coronavirus family that infects humans, bats, and camels. An enveloped, positive-sensed, and single-strand RNA virus invades the host cell by binding to the dipeptidyl peptidase 4 (DPP4) receptor [31]. It is known for causing Middle East Respiratory Syndrome. Twenty-seven counties have reported cases of MERS since 2012. Symptoms of MERS include cough, fever, shortness of breath, and sometimes pneumonia. Some gastrointestinal symptoms such as diarrhea are also observed. It is severe and sometimes fatal for people with weak immune systems, old age, and patients with other serious health conditions such as chronic lung disease, cardiovascular disease, and diabetes. It has a mortality rate of 35%. The major reservoir host for MER-CoV and source of infection from animals to humans are dromedary camels. The transmission route for humans is through close contact with infected individuals (WHO). The spike protein (S) found in the MERS-CoV is a type one membrane fusion protein that can serve as the main structural protein for the virus. To reach the target cell, it binds to the DPP4 receptor [32]. In one study, a DNA construct carrying the spike gene is cloned in the pcDNA3.1 mammalian expression vector is used. It is then transformed into genetically modified *E. coli* bacteria DH10β to prepare OMVs that contain the spike protein; this new OMVs-based vaccine candidate was immunized to BALB/c mice; they were evaluated by comparing it with vaccinated mice with empty OMVs and inactivated MERS-CoV vaccine as control groups. The result showed that vaccinated mice develop potent neutralizing antibodies (nABs) titer against MERS-CoV compared to the inactivated virus controls after eight weeks of vaccination [32]. In another approach, an OMVs-based dual vaccine (OMVs-H1/RBD) for Influenza A virus (H1N1) and MERS-CoV was engineered by producing OMVs using an antigenic chimeric fusion protein of H1-type haemagglutinin (HA) (containing both HA1 and HA2) of the pandemic H1N1 strain from 2009 (H1N1pdm09) and the RBD of the MERS-CoV. It was observed that the chimeric antigen induces specific neutralizing antibodies (nABs) against both the strains resulting in the protection of immunized mice against H1N1pdm09 and effective neutralization of MERS-CoV. The MERS-CoV spike protein (S) plays an important role during the entry of the virus via the binding of its antigenic RBD region to the DPP4 host cell receptor. The RBD is an antigenic glycoprotein fragment that induces humoral and cellular neutralizing antibody (nAbs) immune responses [33]; these studies show that OMV-based vaccines provide a reliable and safe approach to protecting against viral infections (Figure 4). 

Notably, MERS-CoV-associated S protein shows 74% homology with SARS-CoV-2 S protein. Rabets et al., reported that due to higher sequence similarity in S protein of MERS-CoV and SARS-CoV-2 it shows the cross-reactivity against the spike protein of each other [34]. 

Some vaccines are developed, and many are under trial to sustain this pandemic. Two OMV-based vaccine candidates with protein subunit platforms are under pre-clinical or phase 1 trial by Quadram Institute Biosciences, BiOMViS Srl/Univ. of Trento. A vaccine candidate with OMV subunit and protein subunit platform is under trial by Intravacc/Epivax. Another OMV based vaccine with peptide adjuvant and protein subunit platform is under trial by Intravacc/Epivax. [35]. In a recent preliminary study, a novel platform for the vaccine is proposed using the recombinant receptor-binding domain (rRBD) from spike (S) protein of SARS-CoV-2 and OMVs from *N. meningitides* along with aluminum hydroxide as an adjuvant and immunized to animals. Increased IgG production was observed in sera, after 37 days of immunization IgA production was also observed. An increase in IFN-y-producing cells was observed in splenocytes, suggesting that bacterial-induced lethality occurs through TH1 and TH2 cell responses. Thus, immunization with rRBD plus OMVs shows a promising platform for the development of the COVID-19 vaccine [36].

## 5. Preparation and Separation of OMVs

The bacteria are initially grown in a culture medium; they are subjected to centrifugation, and OMVs are collected in the supernatant and filtered through a 0.45-micrometer pore-size filter to separate any residues; they are then subjected to ultracentrifugation at 100,000 to 200,000 rpm and are pelleted. The ultracentrifugation conditions can be adjusted to have a more uniform collection of outer membrane vesicles. Sequential density gradient centrifugation can enhance the purity and quality further, and an inert medium is prepared using substances such as iodixanol, sucrose, dextran, and the OMVs get precipitated about a specific position in the gradient when subjected to ultracentrifugation (Figure 5). Further purification techniques might have to be carried out to remove structures such as flagella, pili, etc. Quantification can be carried out using Nano sight or flow cytometry [37]. NanoSight enables the characterization of particles from 10–2000 nm in solution [38]. With the help of this method, bacterial OMV of *E. coli*, *P. aeruginosa*, and *Salmonella typhimurium* have been separated. [25]. After centrifugation, adding the saturated salt solution to the supernatant will allow the OMVs to be precipitated out. Homogenized salt solution with high solubility and ion strength such as ammonium sulfate is preferred. The salt concentration increased when used to precipitate the OMVs out; this is attributed to the proteins from the supernatant being bound to the salt [26]. Gel filtration can also be used to separate OMVs. Here, the filtration technique is based on separation based on the porous network structures. 

Major disadvantages associated with the precipitation technique would be to prepare the salt solution with precise concentrations. OMVs obtained from all methods listed above will have to be purified [25]. Production is followed by purification. Purification of native OMVs is carried out by tangential flow filtration (TFF). The first round separates the bacteria supernatant with OMVs, and the second round separates other impurities from the OMVs [27]. dOMVs are usually of different sizes thus they are first subjected to sonication to make them smaller and uniform, they may undergo some mutations to supplement their endotoxicity and are then purified. mdOMVs have a high endotoxicity, which is first reduced by modifying the structure of lipid A resulting in the reduced stimulation of Toll-like receptor (TLR) in host cells [28]. Bacteria have very specific glycans [39], while macrophages prioritize particles with “foreign” glycans [40] and this also can be a beneficial factor for OMV as stimulators of the immune response.

## 6. Uptake of OMVs by Host Cells

OMVs can enter through many ways: macropinocytosis, lipid raft-dependent or independent endocytosis, clathrin, caveolin, and dynamin-dependent entry [29]. LPS is usually delivered through endocytosis; O antigen structural region is crucial to OMV entry; if OMVs lack antigen, they can use clathrin-dependent endocytosis to enter the host cell. PAMPs facilitate TLR signaling to facilitate OMV entry into host cells [30]. OMVs mimic the pathogenicity of bacteria; these invade the epithelial lining of the host cells and present themselves to the body’s immune cells, such as neutrophils, macrophages, and dendritic cells in the submucosa, thus activating the immune response. B and T lymphocytes will also be stimulated, thus, enabling a comprehensive immune response. When treated with *E. coli*, OMVs can also cause apoptosis of host cells by developing the G2 phase arrest, or they can carry virulence factors that cause cell death of epithelial cells of the gut [34]. *Neisseria meningitidis* OMVs can stimulate the human neutrophils to produce cytokines and chemokines such as interleukin 1-beta, IL-8, tumor necrosis factor-alpha, macrophage inflammatory proteins 1 alpha, and 1 beta [41]. Gamma interferon-stimulated can maintain or increase the inflammation reaction. *E. coli* OMVs have cytotoxic necrotizing factors 1; these can reduce the membrane fluidity of polymorphonuclear leukocytes, thereby decreasing the levels of cytokines and chemokines [42]. OMVs make macrophages secrete proinflammatory substances such as chemokines and cytokines; they are phagocytosed by macrophages, activating them, then induce other immune molecules such as interleukin 1-beta, IL-8, tumor necrosis factor-alpha, and macrophage inflammatory proteins 1 alpha and 1 beta [38]. *Legionella pneumophila* OMVs facilitate the replication of the pathogen in the host macrophages [43].

OMVs can also induce macrophage remodeling leading to dysfunction of immune cells. OMVs can play an important role in secreting anti-inflammatory molecules such as IL 10. *Porphyromonas gingivalis*, *Treponema denticola*, and *Tannerella forsythia* OMVs phagocytosed by macrophages; they also stimulate the macrophages to release immune molecules such as TNFα, IL-8, and IL-1β and activate the NF-κB complex; they also primed and activated the inflammasome complex [44]. A study found that the morphology of the macrophages of *E. coli* OMV immunized mice was changed to a spindle-like shape, showcasing how OMVs activate macrophages. OMVs induce Th1 and Th17 immunity, producing macrophages that are much more effective at reducing bacterial load [39].

*Aggregatibacter actinomycetemcomitans* bacteria that cause periodontal disease carry extracellular RNA that stimulates the human macrophages to produce TNF-α via the TLR-8 and NF-κB signaling [40]. *Shigella* OMVs stimulate macrophages to secrete both proinflammatory and anti-inflammatory (IL-12, IL-18, IL-6, and IL-10) molecules; IFN-γ stimulates the inducible isoform iNOS in macrophages. It was seen that multi-serotype OMVs produce a higher immune response than single-serotype OMVs [45]. Thus, OMVs can stimulate a strong immune response yet save bacteria by protecting them with anti-inflammatory molecules [46]. Meningococcal OMVs stimulate dendritic cells via high expression of CD40, CF83, CD80, CD86, and IL-6 1L1-beta [47]. *Helicobacter pylori* enhance dendritic cells to express Heme oxygenase 1 (HO-1). Cumulatively, these simulations will have a role to trigger the adaptive immune response [48]. Maturation of human monocyte-derived DCs, murine bone marrow-derived DCs, and CD11c+ splenic DCs were seen to be induced by *Salmonella typhimurium* OMVs. In addition, they didn’t cause pyroptosis of the cells but rather helped increase the life span; they also enhanced the cross-presentation BMDCs and splenic CDC11c+ DCs to OTI CD8+ T cells, dependent on MyD88.

There are numerous therapeutic benefits for this DC-mediated cross-presentation leading to enhanced CD8+ T cell response; this response was dependent on the LPS content of the OMVs and a downstream adapter protein myeloid differentiation primary response 88 (MyD88) involved in TLR signaling [49]. *Bacteroides thetaiotaomicron*, a common bacterium present in the intestinal tract, produces OMVs that can cross the endothelial lining of the intestine and stimulate the dendritic cells to produce IL10, an immunoregulatory cytokine that is important for maintaining intestinal homeostasis [50]. 

*Salmonella typhimurium* OMVs activate both B and T lymphocytes, however, they can also directly activate the B lymphocytes [51]. *Neisseria lactamica* can induce polygonal IgM via B cell proliferation increasing the immune tolerance and colonizing the host [52]. *Morrell cartarrhalis* and *Haemophilus influenza* use OMVs to help escape the host’s immune system. IgG and IgM produced by B cells cannot recognize the pathogen [53]. The mechanism by, which B cells respond to OMVs is elucidated: IgD and B cell receptors cluster and calcium ions are mobilized. Other molecules essential for activation are triggered, such as IgD binding superantigen and unmethylated CpG DNA; this then leads to the production of other immune-active molecules such as IL6 and IgM followed by activation of CD19+ and IgD+ lymphocytes, creating a wide range of effects on the B cells [54]. 

The humoral immunity is activated, and Th1, Th2, and Th17 differentiate and contribute to the immune response. In addition, CD4+T and CD8+T cell responses are also triggered by OMVs [55]. OMVs can also suppress T cell response, *Neisseria meningitides* can negatively impact T cell production by changing the receptors [46]. Proteins on the OMVs bind to human carcinoembryonic antigen-related cell adhesion molecule 1 (CEACAM1) on CD4+ T cells and inhibit them. *Neisseria gonorrhoeae* Por B can stop CD4+T cell proliferation and alter immune suppression [55]. OMVs of *H. pylori* can suppress immune response through apoptosis (Figure 6).

## 7. OMVs-Based Therapeutics

The production of OMVs is very cost-effective, and it is quite easy to scale up the production of OMVs in inexpensive liquid media. OMV production can be increased by inducing hyper-vesiculation in the bacteria. Notably, hyper-vesiculation-based increased production of OMVs was successfully done in *Neisseria meningitidis*. OMVs are quite stable under various conditions due to the protective lipid outer membrane. *N. meningitidis* OMVs are shown to retain their antigenicity and enzymatic activity while stored at 4 °C. However, storage at 37 °C for three months led to a certain loss in antigenicity [56]. Phosphotriesterase, upon packaging into OMVs of *E. coli* retained enzyme activity 100 times more than free phosphotriesterase [57]. Notably, OMVs maintain the activity of phosphodiesterase even after multiple cycles of freeze and thaw.

## 8. Native OMV Vaccines

The benefits of OMVs in vaccines can be increased multiple folds by necessary modifications genetically. Meningitis B vaccine study has also proved safe and effective by engineering the adverse effects of lipopolysaccharide. The simultaneous reaction of multiple antigens/important antigens and heterologous antigens promotes their application extensively [58]. Natural OMVs are purified and concentrated, and the detergent-based extraction of OMVs helps to reduce the toxicity of the LPS complex. The term native OMV is used to describe intact OMV generation from cell supernatant and for concentration OMVs from dead cells using detergent-free disruption methods [59]. Currently, there are vaccines comprising capsular polysaccharides coupled to a carrier protein for several serogroups of *N. meningitides*; however, this is not possible for serogroup B because it mimics the molecular structures in the brain, thus leading to greater risks of autoimmunity [60]. Polysaccharide combined with antigen becomes immunogenic in infants and prime for memory anticapsular antibody response. To combat these meningococcal OMV vaccines have been developed using DOC detergents to release the OMVs, this also had the added benefit of reducing the toxicity of the LPS complex, however, there was the loss of certain important outer membrane lipoproteins and increased contamination with the inner membrane proteins and cytoplasmic contents [58]. However, this method has been useful to manufacture vaccines against epidemics, where a strain was produced matching the circulating strains and monovalent detergent extracted OMVs were produced from this strain; this vaccine has an efficacy of 83% in Cuba, up to 87% in Norway, and a minimum of 70% in New Zealand in combating meningitis B disease [61]. 

The immune response in infants and children is largely targeted against surface accessible loops on a porin protein called Por A. Thus, these OMV vaccines are effective against one particular strain. To combat multiple strains OMV vaccines must be prepared from more than one strain of mutants expressing more than one type of Por A molecule. However, there are more than 20 different Por A molecules in many outbreaks, making it difficult to create an effective OMV vaccine. However, since the adult immune response is wider than that of children, an OMV vaccine can still be effective in adults [62]. A vaccine was created in the Netherlands, having genes coding for 6 subtypes of Por A molecule. The MenPF1 vaccine against MenB consisting of dOMV is proven safe, and able to produce high bactericidal titers in a phase one clinical trial [63]. Another dOMV vaccine against the serogroups A and W show high serum bactericidal assay and opsonophagocytosis assay compared to unconjugated and conjugated polysaccharide meningococcal vaccines in mice [64]. Recently, mdOMV vaccines have been developed that offer higher nasopharyngeal colonization and induce a broad-spectrum bactericidal immune response against multiple strains. Two more vaccines of mdOMV are under clinical trials, one with deleted capsules of lpxL1 and gna33 and overexpressed fHbp v.1 derived from the African *N. meningitides* W strain [65]. Another has, in addition to the modifications mentioned above, deleted capsules of lpxL1 and gna33 and over-expressed fHbp v.1 has a stabilized expression of Opc A and expression of a second Por A [66].

In addition to serving as vaccines against meningococcal pathogens, OMVs also act as vaccines for other pathogens such as the *Shigella flexneri*; this vaccine could protect mice against shigellosis. With enhanced protection when nOMVs were encapsulated in poly nanoparticles. *Shigella sonnei* mdOMV is in clinical trials and can elicit anti-LPS O antigen-specific antibodies in healthy adults. nOMV vaccines derived from *S. typhimurium* protected against various serotypes due to antibodies being elicited against the outer membrane proteins. mdOMV vaccines produced high anti-O antigen-specific IgG responses, which were much more diverse when compared to those induced by glycoconjugates [67]. Demand for the *Bordetella pertussis* vaccine has grown over the past few years with an increased incidence of whooping cough; this has been associated with the change from whole-cell vaccines to acellular pertussis vaccines, which consist of few antigens associated with alum; though there has been a reduction in mortality rates, immunity was not provided against circulating strains of the bacteria [68]. Whole-cell pertussis vaccines induce Th1 and Th17 responses; acellular pertussis vaccines produce strong antibody but weak Th1 and Th17 responses. It is seen that whole-cell pertussis vaccines induce CD4 T memory cells that are resident in the lungs and are more effective at inducing long-term immunological memory against pertussis. It has been found that OMV vaccines have a better protective function against a PRN (-) strain than acellular vaccines. In addition to this, OMV vaccines also CD4 T cells with a tissue-resident memory (TRM) cell phenotype (CD44+CD62LlowCD69+ and CD103+) were present in the lungs of the mice that were immunized. CD4 TRM cells that release IFN-γ and IL-17, which are known to increase the adaptive immunity against pertussis, are also present in the lungs of mice injected with the OMV vaccine. Thus, the OMV vaccine has a greater immunological effect than the acellular pertussis vaccine and lacks the toxicity of the whole-cell pertussis vaccine making it the ideal vaccine candidate [68]. 

Zika virus wreaked havoc in Brazil and Latin America, presenting itself as a rash febrile illness, associating itself with conjunctivitis, headache, Guillain-Barre syndrome, and microcephaly of the fetus when pregnant women are affected. *N. meningitidis* was grown at 37 °C under 5% of carbon dioxide in agar. Nonetheless, 70% of confluent C6/36 cells with ZIKV showed 75% changes in cellular morphology. OMVs were first added to the C6/36 cells infected with ZIKV and were allowed to fuse with ZIKV; these were then collected from the supernatant. Two formulations of the ZIKV vaccine were created conjugated and not conjugated with silica as an adjuvant. Both the formulations showed significant production of IgG titers, analyzed using ELISA, and were much higher than the control OMV formulation, indicating the vaccine formulations’ efficacy. The mice expressed IL-2, IL-4, and TGF beta chemokines. Thus, both TH1 and TH2 immune response is greater when inoculating with the ZIKV vaccine formulation. Serum collected from inoculated mice was shown to neutralize ZIKV in vitro, determined with real-time PCR [69].

## 9. Heterologous Vaccines

There are two ways by, which the antigenic content of OMVs can be enhanced. One is by inducing mutations so that the proteins present on the outer membrane are not released. The second is by introducing antigens onto the membrane by using autotransporters [58]. Surface-associated antigens would bind with B cells, and luminal antigens can trigger the cytotoxic T-cell response. Thus, depending upon the desired immune response, the OMV can be designed. In a study, the *Salmonella enterica* serovar Typhimurium strain was constructed that secreted a pneumococcal protein PspA. Vesicles secreted from this strain were used to vaccinate mice intranasally. It was seen that mice immunized with the OMVs had serum antibody response Psp A. However, if they were vaccinated using purified Psp A, they lacked an immune response; this showed that OMVs could be used to deliver antigens from a Gram-positive organism and trigger an immune response [70]. Periplasmic alkaline phosphatase PhoA of *E. coli* was introduced into the OMVs of Vibrio cholerae; intranasal immunization in mice led to immunity being developed against this heterologous antigen. In addition to inducing a long-lasting immune response in the female mice, it also conferred immunity to the neonatal offspring [71].

In addition to luminal antigen loading, surface-bound antigens can also elicit a strong immune response. Lipoprotein OspA in *Neisseria meningitidis* is a surface-exposed antigen. The surface-exposed antigen elicited a more robust OspA-specific antibody immune response against outer membrane proteins and LPS than the luminal antigen [70]. O antigen from *Francisella tularensis* has resulted in O antigen-specific IgG production in mice, protecting them against the lethal strain. Heterologous proteins can also be attached post OMV production by a spy catcher coupled with a protein attached to the spy tag [72]. OMVs activate pathogen-associated molecular patterns, including TLR2, TLR5, TLR 9, and TLR13, which recognize lipoproteins, flagellin proteins, and unmethylated CpG motifs; these then stimulate the corresponding Pattern associated Receptors on Antigen Presenting Cells. Human kidney cells have been transfected by expressing TLRs from *Shigella*, *Salmonella*, and *Neisseria meningitides*, mdOMVs from these bacteria have been modified to remove acyl chains from the LPS to reduce endotoxicity. Most of these trigger TLR2, TL4, and TLR5 to induce an immune response. The contribution of the other TLRs to stimulate the Pathogen Receptor cells is low [28].

## 10. OMV-Based Novel Adjuvants

In contrast to the conventionally used synthetic particulate systems, the OMV adjuvant vaccines represent a unique system where both the antigen and delivery vehicle are derived from the pathogen itself. Owing mainly to their size and safety profile in humans, OMVs are regarded as an attractive adjuvant for vaccine delivery [73].

### 10.1. Mechanisms

OMV as an adjuvant is based on Pathogen Associated Molecular Pattern (PAMP), induction of dangerous molecules, and geographic concept. The PAMPs activate both the Pattern Recognizing Receptors (PRRs) and Toll Signalling Receptors (TLRs); these then recruit the cells into active immunity and stimulate the Antigen Presenting Cells (APCs). It is thus thought that OMVs could increase antigen uptake, cell surface expression, and immunostimulatory molecules, aiding in T cell production [74]. Danger molecules affect host cells, leading to mature T cells engaged in an enhanced immune response [75]. The geographic concept involves diverting antigens from the injection site to the tissue-draining lymph node by dendritic cells. Thus, OMVs have immunogenic properties, act as carriers, and show an inherent adjuvant effect [76].

### 10.2. Applications

*Meningitidis* MenB OMVs was used as an adjuvant with group A *meningococcal* capsular polysaccharide. Unlike other adjuvants causing hypersensitivity, these vaccines had low toxicity and elicited a strong T cell response. [77]. OMV derived from *E. coli* can stimulate the humoral and cell-mediated immunity mediated via the IFN-g and IL-17 T cell-dependent response production [78]. In HIV, in addition to inducing IFN-g and IL-17, T cell-dependent response and performing Th1 oriented response. Other particles such as Virus-Like Particles (VLP) induced a high anti-HIV IgG production by using OMV vaccines [79,80]. OMVs have been explored as an adjuvant by mixing them with other known antigens such as keyhole limpet hemocyanin and ovalbumin in the hepatitis B vaccine, triggering an enhanced immune response in the host cells. Lipoprotein OspA in *Neisseria meningitidis* is a surface-exposed antigen. The surface-exposed antigen elicited a more robust OspA-specific antibody immune response against outer membrane proteins and LPS than the luminal antigen [81]. 

### 10.3. Advantages

OMVs trigger both the humoral immune system and cell-mediated immunity, thus providing a comprehensive immune response. Many infectious pathogens affect the mucosa tissue; thus, vaccines targeting the mucosa tissue are quite beneficial. The existing adjuvants fail to deliver antigens to the mucosal tissue; OMV-based vaccines have proved promising in this aspect. OMVs are absorbed into the epithelial cells; they have then mediated through a lipid raft-dependent endocytic pathway and are directed for sorting into lysosomal compartments [82]. Thus, it functions as mucosal transporters carrying antigens that stimulate the APCs for an immune response. Furthermore, it was proposed that the LPS complex of OMVs could direct the B lymphocytes. Thus, mucosal OMV adjuvants have the dual benefit of being a transporter and inducing the immune response [76]. OMVs are uptaken by immune cells and present a range of surface-exposed antigens in native conformation. Moreover, TLRs activating components, represent an attractive and powerful vaccine platform and potentially induce humoral and cell-mediated immune responses.

Relying on flexible genetic modifications, OMVs can also perform several functions by carrying different small molecules with numerous functionalities [83]. Pre-Existing Antibodies do not inhibit the adjuvanticity of OMVs. Despite the presence of OMV-specific antibodies, they could still drive cellular immune responses to a co-delivered peptide [84]; this shows the potency of OMVs as an ideal adjuvant compared to conventional ones. Incorporating OMV adjuvants into new or existing vaccines could catapult the magnitude and breadth of adaptive immune responses to increase the overall vaccine efficacy. This will also represent a hopeful and pioneering development in the field of nanotechnology and vaccinology [84]. 

### 10.4. Limitations 

One of the main factors that OMVs can act as adjuvants is the LPS complex. However, there are concerns associated the LPS toxicity; hence mechanisms to reduce the content of LPS present have been developed. Thus, endotoxins must be removed post-production artificially, but OMV formulations having deficient LPS complexes are less immunologically active. A fine balance must be maintained to maintain the desirable low toxicity and high immunogenicity of the LPS complex. As the formation of OMVs is not explicitly explained, there is some concern with the consistent production of OMVs. Many surfactants could affect OMV integrity, and other factors such as temperature, pressure, and absorption of phages could also affect OMV production. There is also a need for more human and comparative animal trials to prove the efficacy and safety of these novel adjuvants [76].

## 11. Future Prospective and Scope

OMV-based vaccines ignite the hope for the next-generation vaccine’s development. Meanwhile, there are various challenges such as yields of OMVs, isolation process, immunomodulatory and cytotoxicity in the development of OMV vaccines. Naturally, OMV produced by bacteria in very low yield helps bacteria avoid stress and helps in bacterial quorum sensing mechanisms. Few solutions, which elevate the yielding of OMV have been practiced, such as stress, temperature, pressure, and nutrient depletion. Treatment of *Pseudomonas putida*, *Pseudomonas aeruginosa*, and *Bacillus pertussis* with higher temperatures (55–65 °C) increases the release of OMVs [85,86,87]. Significantly, treatment with higher temperature does not interfere with antigen structure; hence delivery of OMVs induces immunogenicity. OMVs stability varies at a temperature ranging from 40 °C to 50 °C depending on their mode of formation. Heat-induced OMVs contain an elevated amount of lysophospholipids and induce comparable immune responses to spontaneous OMVs in vitro. Although various studies reported the antimicrobial activity of antimicrobial peptides, recently, immunomodulatory functions of these peptides have also been determined [88,89,90,91,92]. Few AMPs such as cathelicidin (LL-37) instruct the differentiation of the dendritic cells to evoke proinflammatory responses, an essential function of vaccine candidates for their efficacy. 

Meanwhile, few AMPs induce chemokines as a part of the immunomodulatory response. Besides modulation of proinflammatory responses and chemokine production, LL-37 and other chicken peptides (CATH-2) have been reported for their TLR-4 inhibitor activity to maintain the homeostasis between pro and anti-inflammatory responses. Synthetic anti-endotoxin in formulation with OMVs is also known to alleviate human macrophages’ activation. Hence, OMVs in formulation with AMPs are promising futuristic vaccine candidates. Moreover, tailoring OMVs and AMPs formulation could be a landmark to evoke the desired immune response against the specific disease.

## 12. Conclusions

Over the past few decades, plenty of advancements in vaccine production and its efficacy have been made; for example, traditionally, undefined attenuated or inactivated vaccines are in great use, which slowly change into well-characterized subunit vaccines that resemble resembling the pathogen such as structure in practice. The emerging OMV-based vaccine contains various imperative features for the native configuration of antigens to initiate humoral immune surveillance, which potentially generates a lymphocytic immune response; it also includes the pathogens-associated molecular patterns (PAMPs), which helps in triggering the innate immune response, and size of antigen for their processing through antigen-presenting cells. Moreover, for the successful application of OMVs as a novel platform structure for vaccine development, OMVs must be optimized through appropriate innate and humoral responses after adding or removing any antigenic part from it importantly, whether antigens exposed on the surface or merely present inside the OMV structure is still subject for further research. The development of OMV vaccines based on their complete structural and functional information and scientific understanding paved the way for treating the disease with more efficacy with prolonged immunity.

OMVs are an emerging and promising platform for vaccine development, especially in non-regenerative and acellular vaccines. OMVs-based vaccines are more immunogenic in comparison to non-regenerative and VLP vaccines. OMV-based vaccines are safer than the whole pathogen attenuated vaccine as OMVs do not have the self-replicative capacity, and there is no evolutionary escaping. OMVs vaccines are potential candidates where whole-cell treatment approaches are not applicable. Moreover, the cons of OMV-based vaccines are their low yields and endotoxic effects. Notably, there is continuous improvement in extraction methods of OMVs to increase vesicle formation and genetic modifications to avoid possible endotoxicity. Various factors are responsible for elevated spontaneous vesicle formation in bacteria, such as temperature, stress, and antibiotic treatments. Host-associated antimicrobial peptides have immunomodulatory potential; they act by directing and evoking the proinflammatory (Th1/Th17) response. Importantly, OMV-based vaccines are a potential platform for vaccine development against various bacterial infections. Conclusively, OMV-based vaccines are the future of antibacterial vaccine development.

## Figures and Tables

**Figure 1 vaccines-10-01578-f001:**
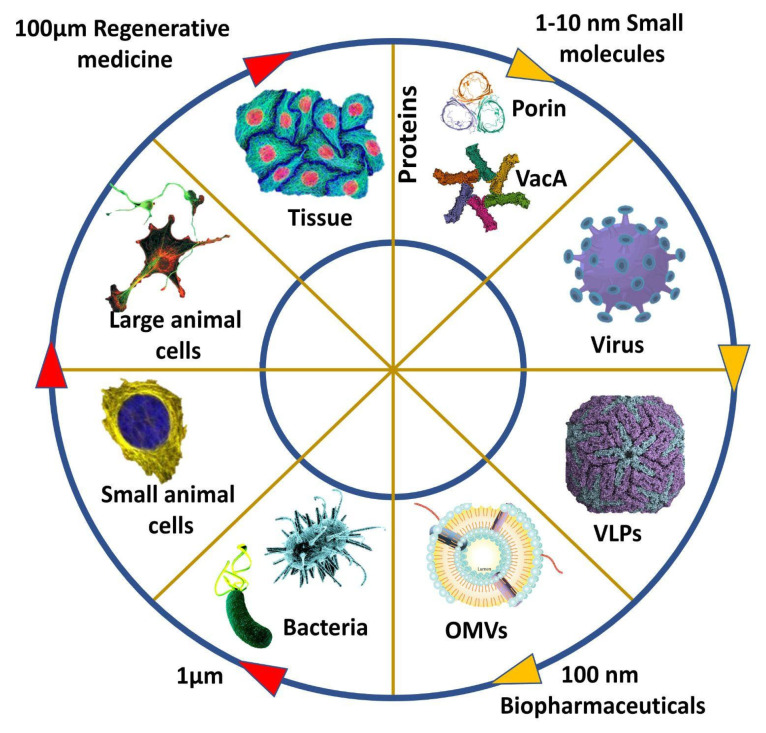
A vaccine as therapeutic candidates from well-defined small pharmaceuticals protein to OMV and large undefined regenerative medicines.

**Figure 2 vaccines-10-01578-f002:**
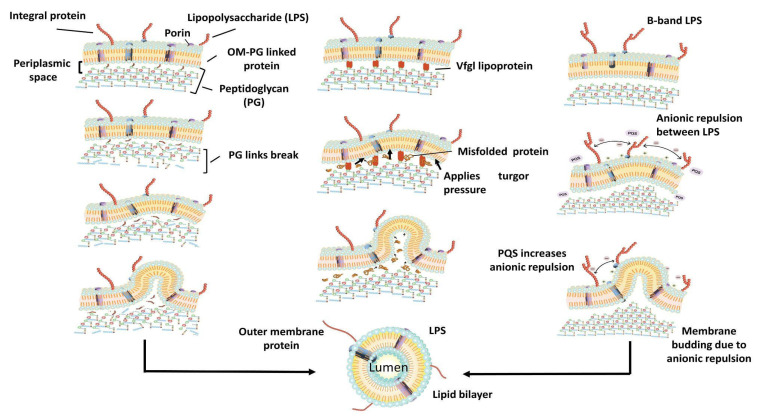
Formation of OMVs. It shows loss or relocation of covalent linkages between the peptidoglycan layer and the outer membrane, peptidoglycan fragments and misfolded proteins protrude into the periplasmic space exerting a turgor pressure on the outer membrane leading to the OMVs pinch-ing off, and the enrichment of the curvature-inducing molecules such as B-band lipopolysaccharide and the quinolone PQS of *Pseudomonas aeruginosa*.

**Figure 3 vaccines-10-01578-f003:**
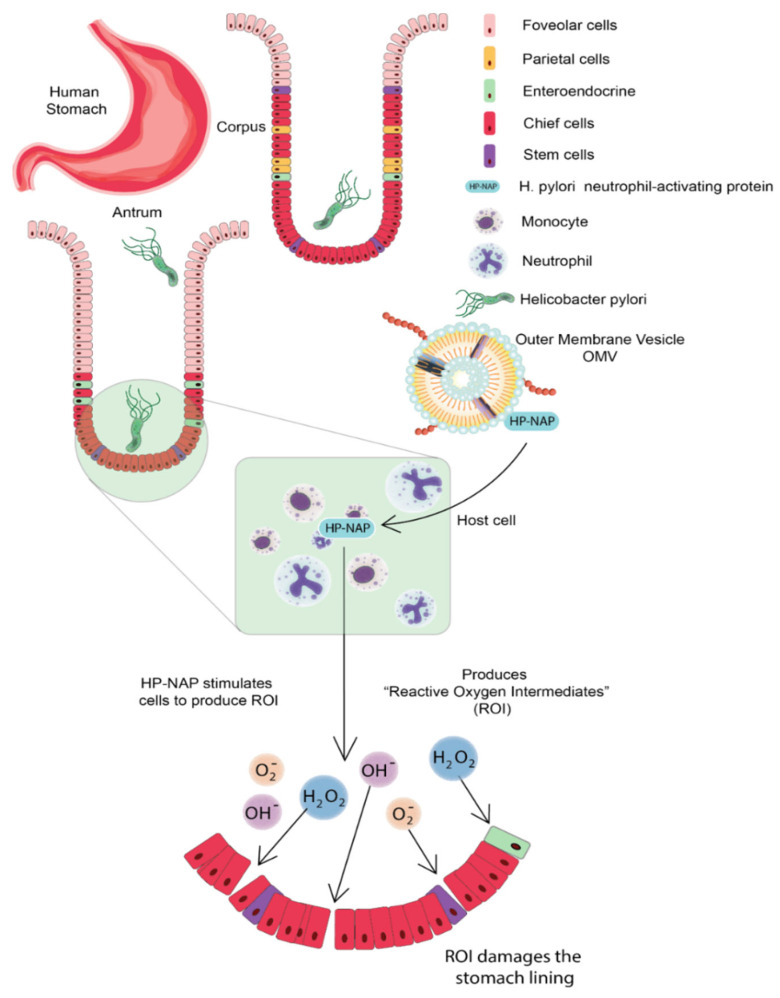
*Helicobacter pylori* Neutrophil Activating Proteins (HP-NAP) stimulates the release of Reactive Oxygen species (ROS), which damages the gastric epithelium. The delivery of HP-NAP through OMV to gastric mucosa thus increases the nutrient availability to bacteria via ROS-mediated mucosal damage. Gastric Epithelium Lining Figure Inspiration from [17].

**Figure 4 vaccines-10-01578-f004:**
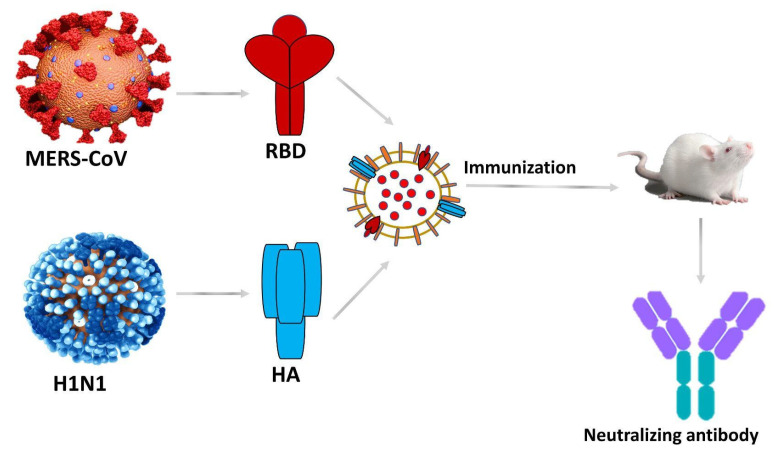
Tailored OMV-based vaccine development against the MERS-CoV and H1N1.

**Figure 5 vaccines-10-01578-f005:**
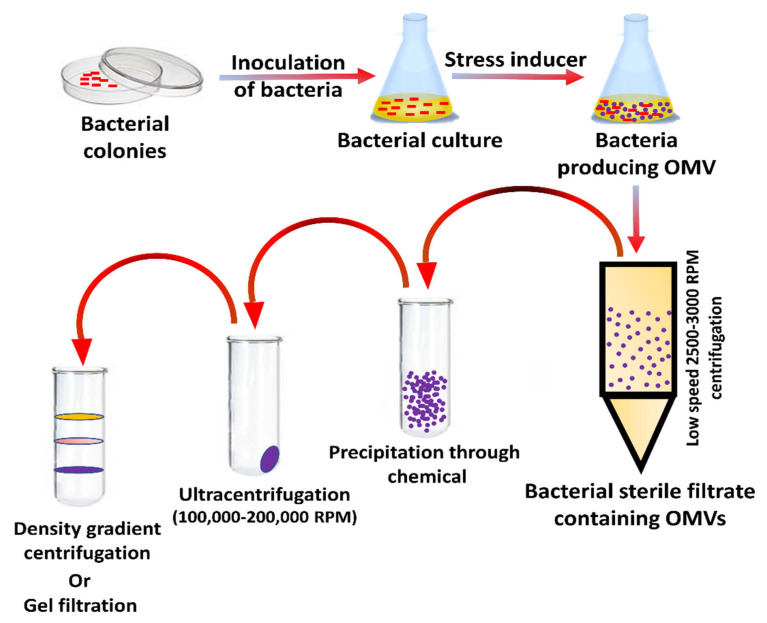
Preparation and separation of OMVs.

**Figure 6 vaccines-10-01578-f006:**
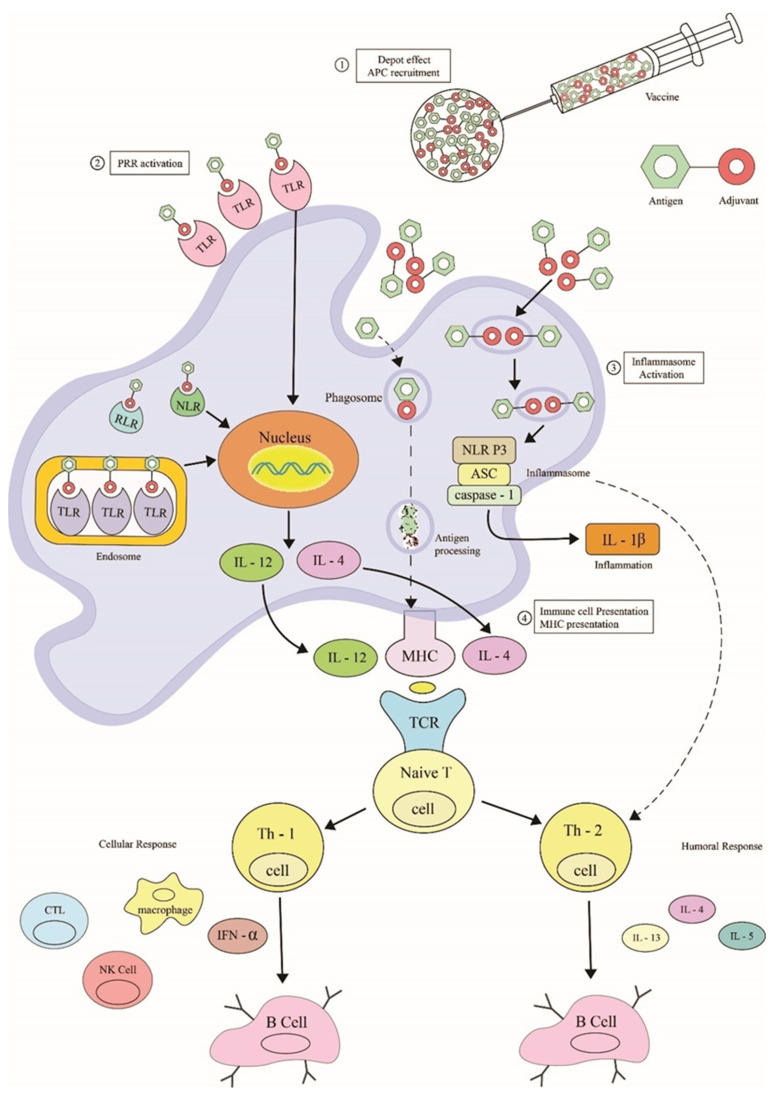
Activation of cellular and humoral response through the OMVs. Legends: APC: Antigen Presenting Cells; PRR: Pathogen Recognition Receptor; TLR: Toll-like receptor; NLR: Nucleotide oligomerization domain (NOD)-like receptor; RLR: Retinoic acid-inducible gene I (RIG-I)-like receptor; IL: Interleukin; MHC: Major Histocompatibility Complex; TCR: T-cell receptor; Th: Helper T-cells; IFN: Interferon; CTL: Cytotoxic T lymphocytes; NK: Natural Killer.

## Data Availability

Data sharing not applicable.

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
