# Peer review of "Outer Membrane Vesicles: An Emerging Vaccine Platform"

_vaccines, 2022, doi:10.3390/vaccines10101578_

Round 1
Reviewer 1 Report
The review article titled “Outer Membrane Vesicles: An Emerging Vaccine Platform” by Kashyap et al., is an attempt to provide a cumulative data of OMVs as a vehicle to deliver immunogens or protective antigens to host. Information provided by authors described the adjuvant properties of OMV based vaccines and their ongoing trials against various pathogens including SARS Co-V-2. The manuscript is well written and backed with published research work. On this, I have just two questions for the authors:
1. The manuscript is focused on OMV as vaccine platform with their adjuvant properties as a prime. However, majority of the published work on OMVs are associated with pathogenic bacteria such as Vibrio, Yersinia, Francisella etc with the objective to achieve a broad spectrum of immune responses against these pathogens. OMVs are not only proving the adjuvant properties but, also negate any fear of infection associated with the whole cell vaccine candidate against such disease causing pathogens. All these informations are missing in the manuscript.
2. Under the sub-title “3. Species producing OMVs”, authors have included recombinant E. coli, and N. meningitidis producing OMVs as promising platform for development of vaccine against SARS CoV-2 and MERS viruses. In my sight this information should be given under a separate sub-title.
Author Response
Response letter
Reviewer1
The review article titled “Outer Membrane Vesicles: An Emerging Vaccine Platform” by Kashyap et al., is an attempt to provide a cumulative data of OMVs as a vehicle to deliver immunogens or protective antigens to host. Information provided by authors described the adjuvant properties of OMV based vaccines and their ongoing trials against various pathogens including SARS CoV-2. The manuscript is well written and backed with published research work. On this, I have just two questions for the authors:
Question:
The manuscript is focused on OMV as vaccine platform with their adjuvant properties as a prime. However, majority of the published work on OMVs are associated with pathogenic bacteria such as Vibrio, Yersinia, Francisella etc with the objective to achieve a broad spectrum of immune responses against these pathogens. OMVs are not only proving the adjuvant properties but, also negate any fear of infection associated with the whole cell vaccine candidate against such disease causing pathogens. All these informations are missing in the manuscript.
Response:
Thank you very much reviewer for your comments to improve the present manuscript. Along with the adjuvant properties, OMVs also provide complete immunity as they carry the antigen of pathogens. Moreover, the non-replicative nature of OMVs makes them advantageous for antigen delivery to the host thus denying any fright of infection associated with whole cell vaccine against disease-causing pathogens. OMVs stimulate the innate immune system of the host via the activation of TLRs and NLRs as they contain various PAMPs like lipoproteins, LPS, and pathogenic DNA fragments 1. Studies have shown that OMVs with attenuated endotoxicity provide enhanced mucosal adjuvant properties in comparison to intranasal vaccines adjuvanted with cholera toxins. These OMVs are less reactogenic in comparison to native ones and also resulted in an increase in the production of IgG and IgA 2. We have added this part in detail and highlighted it in yellow color (Section4, line number 261-300).
Question:2
Under the sub-title “3. Species producing OMVs”, authors have included recombinant E. coli, and N. meningitidis producing OMVs as promising platform for development of vaccine against SARS CoV-2 and MERS viruses. In my sight this information should be given under a separate sub-title.
Response:
Thank you very much reviewer for your comment. As per your suggestion we have shifted the subsection 3.4 and 3.5 MERS-CoV and SARS-CoV-2 respectively into a new section4 OMV Based Vaccine Delivery and we merged MERS-CoV and SARS-CoV-2 according per other reviewer comment (Section4 and subsection4.1)
Reviewer 2 Report
In this review, the authors summarized the function, formation and application of outer membrane vesicles (OMV), highlighting the role of OMV in vaccine adjuvant and design. The contents are suitable for Vaccines. However, there are some major problems to be further improved as well:
1. The part of “6. Advantages of OMVs in therapeutics” could be changed into “OMV-based therapeutics”;
2. The parts of 9, 10, 11, 12 and 13 should be merged into one part “OMV-based novel adjuvants” with subtitles like “9.1 Mechanisms”, “9.2 Applications” and so on.
3. The contents about OMV-based vaccine delivery should be added into the manuscript. There are many recent excellent works focusing on vaccine delivery strategies using OMVs.
4. In section 3 “Species producing OMVs”, the parts of 3.4 MERS-CoV and 3.5 SARS-CoV-2 should be moved to the new part of “OMV-based vaccine delivery” mentioned above.
5. The review needs more pictures or schemes to quickly convey key information, such as in Part 4 “Preparation and separation of OMVs”.
Author Response
Response letter
Reviewer2
Comments and Suggestions for Authors
In this review, the authors summarized the function, formation and application of outer membrane vesicles (OMV), highlighting the role of OMV in vaccine adjuvant and design. The contents are suitable for Vaccines. However, there are some major problems to be further improved as well:
Question:
- The part of “6. Advantages of OMVs in therapeutics” could be changed into “OMV-based therapeutics”;
Response:
Thank you reviewer for your comment. As per your suggestion we have replaced the “Advantages of OMVs in therapeutics” with “OMV-based therapeutics” highlighted in yellow color (Section7)
Question:
- The parts of 9, 10, 11, 12 and 13 should be merged into one part “OMV-based novel adjuvants” with subtitles like “9.1 Mechanisms”, “9.2 Applications” and so on.
Response:
Thank you reviewer for your comment. We have merged section 9 to 13 into a new section 10 followed by subsection 10.1 to 10.4 and highlighted in yellow color.
Question:
The contents about OMV-based vaccine delivery should be added into the manuscript. There are many recent excellent works focusing on vaccine delivery strategies using OMVs.
Response:
Thank you, reviewer, for your comments. As per your suggestion we have added a new section OMV-based vaccine delivery in the main manuscript highlighted with yellow color (Section4, line number 261-300)
Host immune responses are the first line of barrier to stopping any infection, but occasionally immoderate responses could lead to lethargic tissue damage. Thus, there is a need for a system that avoids excessive immune responses. Nonetheless, OMVs play an important role in the delivery of various virulence factors to host cells, which initiate the host immune response. Therefore, OMVs based delivery system may be utilized to educate the immune system to counter pathogens in a controlled manner. Studies by Jang, Elmi, and Bauman showed that OMVs induce the host immune responses via interacting with the host epithelial cells 3–5. Another study also revealed that OMV of H. pylori carrying CagA is sufficient to evoke the immune response of gastric epithelial cells in place of bacteria 5,6. Moreover, OMVs from Campylobacter jejuni contain outer membrane proteins that also potentially induce host immune response in the intestine 4. Nonetheless, various pathogens also evoke a pro-inflammatory response viz; P. aeruginosa cytokines and chemokines production in lung epithelial, and macrophage cells in mice and rodents respectively. OMVs of other bacteria such as N. meningitidis, L. pneumophila, and H. pylori, salmonella spp enhance the proinflammatory response through the production of tumor necrosis factor-alpha (TNF-α), Interleukin-1 beta (IL-1β); chemokines (CXCL8, CCL3, and CCL4) and degranulation of eosinophils 7–9.
OMVs interact with dendritic cells and activate innate and adaptive immunity. OMVs have the potential to interact with a variety of immune cells, hence provoking proinflammatory responses. OMVs have enormous potential as endogenous vaccine candidates. Since OMVs carry surface antigens identical to that of bacterial surfaces it limits the ability of pathogens to mutate all the target antigens present on vaccine thereby decreasing its ability to generate vaccine-resistant mutant. Also, OMVs in their native state possess many immunogenic properties like self-adjuvation and uptake by immune cells. This makes them an excellent autologous cell derived vaccine delivery platform 10. Bioengineered OMV based vaccines can be developed with increased yields and reduced endotoxicity decorated with homologous and heterologous antigens to treat specific diseases. A versatile OMV-based vaccine platform has been developed to elicit an anti-tumor immune response by presenting the antigens onto OMV surface. Decorated with different protein catchers, the OMVs were able to display distinct tumor antigens. An effective tumor vaccine vector displaying neoantigens was thus developed 11.
Question:
- In section 3 “Species producing OMVs”, the parts of 3.4 MERS-CoV and 3.5 SARS-CoV-2 should be moved to the new part of “OMV-based vaccine delivery” mentioned above.
Response:
Thank you very much reviewer for your comment. We have merged subsections 3.4 and 3.5 into a new subsection 4.1 and shifted the contents in section4, highlighted in yellow color (Section4, line number 304-354)
Question:
- The review needs more pictures or schemes to quickly convey key information, such as in Part 4 “Preparation and separation of OMVs”.
Response:
Thank you, reviewer for your comment. We have added one new scheme representing the preparation and separation of OMVs and assigned it figure 5 and added in section 5 highlighted with yellow color.
Reviewer 3 Report
The current manuscript summarizes current state-of-the-art knowledge about OMV and describes their potential use as an immune buster. A manuscript is easy to read, objective, novel, and comprehensive. I like the style how the authors present the recent finding in OMV research.
With the aim of improving the manuscript I would like to suggest a few ideas for authors considerations as well as minor technical errors:
182 - after stating the IL-8 production I would suggest adding the text that this cytokine is acting as a chemoattractant to involve more neutrophils in the site of production.
187 and elsewhere - I would change ROI for ROS (species) as it is a more accepted abbreviation.
266 - Check Pneumonia from capital letter.
260 - 325. It is worth nothing either in 3.4. or 3.5 that MERS and SARS-CoV-2 have many common epitopes, and can induce cross-reactive immune response [10.1007/s00005-021-00607-8], thus data described in 3.4. can be easily extrapolated to SARS-CoV-2.
Data in lines 336-338 is contradicting the later statement in 434 . I still fill that ultracentrifugation and other techniques are not making OMV production available to many research groups/laboratories. But this is my personal impression.
In the paragraph starting from 358 it is worth mentioning that bacteria have very specific glycans [10.1039/c4cc00660g], while macrophages prioritize particles with "foreign" glycans [10.1074/jbc.M111.273144] and this also can be a beneficial factor for OMV as stimulators of the immune response.
415, 417 - remove doi from text
Figure 6 - decode PLR activation and make increase text size. Also, place of 3 - Inflamasome activation is unclear
444 - Nano sight needs explanation, it is probably a TM
452 -check the sentence and read the whole paragraph for clarity.
456 - detergent OMV - needs explanation
461 - italic
464 - Moreover, ... -the sentence is lacking meaning.
470-472 - the disease should be clearly indicated.
522 - 75% cytoplasmic - something is missing there
521 - C6/36 - cells is missing Check this paragraph for clarity.
568 - citation is needed, many statements are not justified. Since Al adjuvant topic is complicated and the main idea is that OMV can be immune busters (=adjuvants) I would suggest removing paragraph 565-578 at all and starting the next chapter from lines 579. This will omit controversies and discussions.
Paragraph 620 - 628 is based on one paper only and maybe more justification would be better accepted.
Merging chapters 9-14 can be beneficial, but this is just a subjective idea. Adding a scheme or graph could also be beneficial.
Author Response
Response letter
Reviewer3
Comments and Suggestions for Authors
The current manuscript summarizes current state-of-the-art knowledge about OMV and describes their potential use as an immune buster. A manuscript is easy to read, objective, novel, and comprehensive. I like the style how the authors present the recent finding in OMV research.
With the aim of improving the manuscript I would like to suggest a few ideas for authors considerations as well as minor technical errors:
Question:
182 - after stating the IL-8 production I would suggest adding the text that this cytokine is acting as a chemoattractant to involve more neutrophils in the site of production.
Response:
Thank you very much reviewer for our comment. As per your suggestion we have modified the sentence and highlighted it with yellow color (Section3, subsection3.1, line number 182-183).
Question:
187 and elsewhere - I would change ROI for ROS (species) as it is a more accepted abbreviation.
Response:
Thank you very much reviewer for your comment. We have replaced the ROI with ROS and throughout the manuscript.
Question:
266 - Check Pneumonia from capital letter.
Response:
Thank you, reviewer for your comment. We have rectified the word Pneumonia and now it looks like pneumonia and highlighted with yellow color in the main text (subsection4.1, line number308).
Question:
260-325. It is worth nothing either in 3.4. or 3.5 that MERS and SARS-CoV-2 have many common epitopes and can induce cross-reactive immune response [10.1007/s00005-021-00607-8], thus data described in 3.4. can be easily extrapolated to SARS-CoV-2.
Response:
Thank you, reviewer for your comment. We have merged subsection 3.4 MERS-CoV and 3.5 SARS-CoV-2 on the basis of similarity and cross reactivity between structural proteins. Notably, MERS-CoV associated S protein shows 74% homology with SARS-CoV-2 S protein. Additionally, Rabets et al., reported that due higher sequence similarity in S protein of MERS-CoV and SARS-CoV-2 it shows the cross reactivity against the spike protein of each other, highlighted in yellow color (Subsection4.1)
Question:
Data in lines 336-338 is contradicting the later statement in 434. I still fill that ultracentrifugation and other techniques are not making OMV production available to many research groups/laboratories. But this is my personal impression.
Response:
Thank you very much reviewer for your comment. We have removed the contradictory line 336-338 and started with the next sentence.
Question:
In the paragraph starting from 358 it is worth mentioning that bacteria have very specific glycans [10.1039/c4cc00660g], while macrophages prioritize particles with "foreign" glycans [10.1074/jbc.M111.273144] and this also can be a beneficial factor for OMV as stimulators of the immune response.
Response:
Thank you, reviewer for your visionary comments. We have added these lines in the same paragraph as you have mentioned and highlighted them with the yellow color (Section5, line number384-386).
Question:
415, 417 - remove doi from text
Response:
Thank you, reviewer for comments we have removed the doi from the line number 415, and 417.
Question:
Figure 6 - decode PLR activation and make increase text size. Also, place of 3 - Inflammasome activation is unclear. Decoding can be written in the figure caption:
Response:
Thank you, honorable reviewer for your comments. We have increased the font size of figure number-6 and also added the decoding in the figure caption. Now it looks like “Figure 6. Activation of cellular and humoral response through the OMVs. Legends: APC: Antigen Presenting Cells; PRR: Pathogen Recognition Receptor; TLR: Toll-like receptor; NLR: Nucleotide oligomerization domain (NOD)-like receptor; RLR: Retinoic acid-inducible gene I (RIG-I)-like receptor; IL: Interleukin; MHC: Major Histocompatibility Complex; TCR: T-cell receptor; Th: Helper T-cells; IFN: Interferon; CTL: Cytotoxic T lymphocytes; NK: Natural Killer”
Question:
444 - Nano sight needs explanation, it is probably a TM
Response:
Thank you very much reviewer for your comment. We have tried to explain the NanoSight in main text (Section5, line number 365-366). To the best of our knowledge the following possible explanation of NanoSight.
Nanoparticle tracking analysis (NTA) is an advanced powerful characterization technique that combines the properties of both laser light scattering microscopy and Brownian motion in order to obtain size distributions of particles in liquid suspension. NanoSight instruments (Malvern, UK), which currently are the most widely used instruments for NTA in the EV field, are equipped with one or more lasers and an optical microscope connected to a digital camera. According to the manufacturer, NanoSight enables characterisation of particles from 10–2000 nm in solution. Particles are visualised by the light they scatter upon laser illumination, and their Brownian motion is monitored. The NTA software enables sizing of single particles by tracking their mean squared displacement and thereby calculating their theoretical hydrodynamic diameter using the Stokes Einstein equation. On the basis of knowing the analysed sample volume, NTA also allows for an estimation of particle concentration.
Question:
452 -check the sentence and read the whole paragraph for clarity.
Response:
Thank you, reviewer for your comments. We have modified the sentences and removed the redundant part from the paragraph. We have also shifted the few sentences to section-4 and highlighted them with yellow color.
Question:
456 - detergent OMV - needs explanation
Response:
Thank you, dear reviewer, for your valuable correction. Detergent OMV in this case refers to detergent extracted OMVs. However, to maintain the current scientific terminology used all while avoiding any confusion, we have changed the term “detergent OMV” to “detergent extracted OMV” (Section8, line nuber484-485).
Question:
461 – italic
Response:
Thank you, reviewer for your comments we have replaced the word (N. meningitides) in italics (N. meningitides) font and highlighted with yellow color (Section8, line number489).
Question:
464 - Moreover, ... -the sentence is lacking meaning.
Response:
Thank you very much reviewer for your comments. We have removed the incomplete sentence and started in continuation with the next sentence.
Question:
470-472 - the disease should be clearly indicated.
Response:
Thank you, honorable reviewer for your comment. We have mentioned the particular disease in place of endemic and highlighted with yellow color (Section8, line number500).
Question:
522 - 75% cytoplasmic - something is missing there.
Response:
Thank you for your comment. We have modified the sentence for clarity and highlighted it with yellow color (Section8, 548-549).
Question:
521 - C6/36 - cells are missing Check this paragraph for clarity.
Response:
Thank you very much for your comment. We have modified the sentence to be C6/36 cells and highlighted with yellow color. Furthermore, we removed the last sentence of this paragraph which was quite confusing.
Question:
568 - citation is needed, many statements are not justified. Since Al adjuvant topic is complicated and the main idea is that OMV can be immune busters (=adjuvants) I would suggest removing paragraph 565-578 at all and starting the next chapter from lines 579. This will omit controversies and discussions.
Response:
Thank you very much reviewer for your comment. As per your suggestion we have removed the sentence from 565-578 and started the next in continuation with the next paragraph.
Question:
Paragraph 620-628 is based on one paper only and maybe more justification would be better accepted.
Response:
Thank you very much reviewer for your comment. We have justified the mentioned information from three different sources cited in main text. Moreover, we have incorporated more information in the main text and highlighted in yellow color. OMVs are uptaken by immune cells and present a range of surface-exposed antigens in native conformation. Moreover, TLRs activating components, represent an attractive and powerful vaccine platform and potentially induce humoral and cell-mediated immune responses (https://doi.org/10.3389/fimmu.2016.00562, https://doi.org/10.3389/fimmu.2022.987419).
Question:
Merging chapters 9-14 can be beneficial, but this is just a subjective idea. Adding a scheme or graph could also be beneficial.
Response:
Thank you very much reviewer for your comment to improve the present manuscript. As per your suggestion we are also thought merging chapters 9 to 13 make sense and beneficial for the reader. Hence, we have merged the mentioned chapters and continue with the next chapter.
References:
(1) Kaparakis-Liaskos, M.; Ferrero, R. L. Immune Modulation by Bacterial Outer Membrane Vesicles. Nat Rev Immunol 2015, 15 (6), 375–387. https://doi.org/10.1038/nri3837.
(2) Holst, J.; Martin, D.; Arnold, R.; Huergo, C. C.; Oster, P.; O’Hallahan, J.; Rosenqvist, E. Properties and Clinical Performance of Vaccines Containing Outer Membrane Vesicles from Neisseria Meningitidis. Vaccine 2009, 27, B3–B12. https://doi.org/10.1016/j.vaccine.2009.04.071.
(3) Jang, K.-S.; Sweredoski, M. J.; Graham, R. L. J.; Hess, S.; Clemons, W. M. Comprehensive Proteomic Profiling of Outer Membrane Vesicles from Campylobacter Jejuni. Journal of Proteomics 2014, 98, 90–98. https://doi.org/10.1016/j.jprot.2013.12.014.
(4) Elmi, A.; Watson, E.; Sandu, P.; Gundogdu, O.; Mills, D. C.; Inglis, N. F.; Manson, E.; Imrie, L.; Bajaj-Elliott, M.; Wren, B. W.; Smith, D. G. E.; Dorrell, N. Campylobacter Jejuni Outer Membrane Vesicles Play an Important Role in Bacterial Interactions with Human Intestinal Epithelial Cells. Infection and Immunity 2012, 80 (12), 4089–4098. https://doi.org/10.1128/IAI.00161-12.
(5) Ismail, S.; Hampton, M. B.; Keenan, J. I. Helicobacter Pylori Outer Membrane Vesicles Modulate Proliferation and Interleukin-8 Production by Gastric Epithelial Cells. Infection and Immunity 2003, 71 (10), 5670–5675. https://doi.org/10.1128/IAI.71.10.5670-5675.2003.
(6) Turkina, M. V.; Olofsson, A.; Magnusson, K.-E.; Arnqvist, A.; Vikström, E. Helicobacter Pylori Vesicles Carrying CagA Localize in the Vicinity of Cell–Cell Contacts and Induce Histone H1 Binding to ATP in Epithelial Cells. FEMS Microbiology Letters 2015, 362 (11), fnv076. https://doi.org/10.1093/femsle/fnv076.
(7) Lapinet, J. A.; Scapini, P.; Calzetti, F.; Pérez, O.; Cassatella, M. A. Gene Expression and Production of Tumor Necrosis Factor Alpha, Interleukin-1β (IL-1β), IL-8, Macrophage Inflammatory Protein 1α (MIP-1α), MIP-1β, and Gamma Interferon-Inducible Protein 10 by Human Neutrophils Stimulated with Group B Meningococcal Outer Membrane Vesicles. Infection and Immunity 2000, 68 (12), 6917–6923. https://doi.org/10.1128/IAI.68.12.6917-6923.2000.
(8) Alaniz, R. C.; Deatherage, B. L.; Lara, J. C.; Cookson, B. T. Membrane Vesicles Are Immunogenic Facsimiles of Salmonella Typhimurium That Potently Activate Dendritic Cells, Prime B and T Cell Responses, and Stimulate Protective Immunity in Vivo. J Immunol 2007, 179 (11), 7692–7701. https://doi.org/10.4049/jimmunol.179.11.7692.
(9) Jäger, J.; Marwitz, S.; Tiefenau, J.; Rasch, J.; Shevchuk, O.; Kugler, C.; Goldmann, T.; Steinert, M. Human Lung Tissue Explants Reveal Novel Interactions during Legionella Pneumophila Infections. Infection and Immunity 2014, 82 (1), 275–285. https://doi.org/10.1128/IAI.00703-13.
(10) van der Pol, L.; Stork, M.; van der Ley, P. Outer Membrane Vesicles as Platform Vaccine Technology. Biotechnology Journal 2015, 10 (11), 1689–1706. https://doi.org/10.1002/biot.201400395.
(11) Cheng, K.; Zhao, R.; Li, Y.; Qi, Y.; Wang, Y.; Zhang, Y.; Qin, H.; Qin, Y.; Chen, L.; Li, C.; Liang, J.; Li, Y.; Xu, J.; Han, X.; Anderson, G. J.; Shi, J.; Ren, L.; Zhao, X.; Nie, G. Bioengineered Bacteria-Derived Outer Membrane Vesicles as a Versatile Antigen Display Platform for Tumor Vaccination via Plug-and-Display Technology. Nat Commun 2021, 12 (1), 2041. https://doi.org/10.1038/s41467-021-22308-8.
(12) Wright, M. Nanoparticle Tracking Analysis for the Multiparameter Characterization and Counting of Nanoparticle Suspensions. In Nanoparticles in Biology and Medicine: Methods and Protocols; Soloviev, M., Ed.; Methods in Molecular Biology; Humana Press: Totowa, NJ, 2012; pp 511–524. https://doi.org/10.1007/978-1-61779-953-2_41.
(13) Dragovic, R. A.; Gardiner, C.; Brooks, A. S.; Tannetta, D. S.; Ferguson, D. J. P.; Hole, P.; Carr, B.; Redman, C. W. G.; Harris, A. L.; Dobson, P. J.; Harrison, P.; Sargent, I. L. Sizing and Phenotyping of Cellular Vesicles Using Nanoparticle Tracking Analysis. Nanomedicine: Nanotechnology, Biology and Medicine 2011, 7 (6), 780–788. https://doi.org/10.1016/j.nano.2011.04.003.
Round 2
Reviewer 2 Report
The manuscript has been sufficiently improved to warrant publication in Vaccines